# Financial Innovation and Financial Inclusion Nexus in South Asian Countries: Evidence from Symmetric and Asymmetric Panel Investigation

**Md. Qamruzzaman [1,\*] and Jianguo Wei [2]**

[1]   School of Business and Economics, United International University, Dhaka 1207, Bangladesh
[2]   School of Economics, Wuhan University of Technology, Wuhan 430070, China; weijg@whut.edu.cn
\*   Correspondence: zaman_wut16@yahoo.com

**Abstract:** This paper examines the nexus between financial inclusion and financial innovation while incorporating financial development and remittance inflows in the case of six South Asian countries—Bangladesh, India, Pakistan, Nepal, Bhutan, and Srilanka—by employing the panel autoregressive distributed lagged model under a linear and nonlinear framework using monthly data over the period 1990M1–2018M12. Further, a Granger-causality test with System GMM specification was performed for assessing directional causality. The study findings from Panel ARDL confirmed the positive association between financial innovation and financial inclusion, which was observed both in the long run and short-run. Considering the nonlinearity in the estimation, the standard Wald test confirms the existence of an asymmetric relationship both in the short-run and in long run horizon regarding causality test results. The study findings support the feedback hypothesis that the presence of bidirectional causality between the financial innovation and financial inclusion is both in the short-run and long run. Since the study findings established a critical relationship between financial innovation and financial inclusion, therefore effective policy guidelines are suggested so that the contribution from financial inclusion and financial innovation can assist in developing a vibrant financial sector.

**Keywords:** financial innovation; financial inclusion; symmetry; asymmetry; ganger-causality

**JEL Classification:** G21; O16; O31

## 1. Introduction

A vibrant financial sector is characterized by diversified financial instruments, efficient financial institutions, a wide range of financial services, and effective integration with economic activities. More specifically, the well-functioned financial sector looks for continuous adaptation, evolvement, and the diffusion of innovative financial assets, institution, and services along with easy access to financial services, and readily available for the population. The role of financial innovation and financial inclusion in the financial system by optimizing financial effectiveness and efficiency. The  role of financial innovation in the financial sector addressed in financial literature such as, assist in improving banking performance (Chipeta and Muthinja 2018), financial efficacy, efficient financial intermediation. On the other hand, the role of financial inclusion also addressed in finance literatures such as the reduction of financing costs (Sarma and Pais 2008), the availability of formal credit, the proliferation of savings (Calderón and Liu 2003; Demetriades and Luintel 1996; Ashraf et al. 2010), quicken the capital formation (Babajide et al. 2015), the bank-based financial institutions development (Swamy 2012) and financial stability. Evidently, the relationship between the financial inclusion and financial innovation

is implied and yet to test in empirical studies. Therefore, this study is an attempt to unveil their existing association and explore their pattern of effect running from each other, that is, symmetry or asymmetry.

Recently, financial issues pertinent to financial inclusion get immense attention among researchers, policymakers, central banks, and financial institutions by admitting its critical role in fostering the financial sector all over the world. Financial inclusion, according to World Bank universal financial access by 2020, is one of the key aspects of poverty mitigation and inclusive economic growth. It is because financial inclusion expedites economic growth through efficient resource allocation, financial efficiency, the reduction of financing costs, lowering information costs for credit approval, and institutional efficiency in managing funds (Sarma and Pais 2008). By acknowledging the nexus between financial inclusion and economic growth, a number of finance scholars including, Kim et al. (2018); Sharma (2016); Sanjaya and Nursechafia (2016); Kamboj (2014); Adeola and Evans (2017) and Babajide et al. (2015) unveiled positive linkages.

Financial inclusion, according to Kumar and Mohanty (2011), is the provision of affordable, accessible and relevant financial products to individual and firms that had previously not been able to enjoy those benefits. Financially included individuals and firms enjoy certain benefits over financially excluded pollution such as smooth income transaction, growing the business with external financing, financial security through savings accumulation, and so forth. In particular, financial inclusion enables the financial integration of the unbanked population into the formal financial system by offering diversified financial services, assets and investment opportunities. Hence, for attracting people in the financial system for enjoying financial services, it is indispensable that financial institutions should expand their financial product and services through the adaptation and diffusion of innovative financial instruments for investment, service for operational efficiency, and the payment mode for intermediation efficiency. Thus, financial institutions persistently seek innovative and improved financial services and assets so that large groups of the population can attract and enable the satisfaction of their needs with innovative financial services and assets in the form of financial innovation. Financial innovation, according to Tufano (2002), is the process of emergence, diffusion, and popularization of new financial instruments, financial institutions, financial technologies, and financial markets in the economy. The presence of financial innovation in the financial system can be addressed in two different wings, such as the product innovation and process innovation. The role of financial innovation in the financial system are multifold which are observed in finance literatures such as, financial services diversification (Silve and Plekhanov 2014; Bianchi et al. 2011), efficient financial intermediation (Johnson and Kwak 2012), technological advancement (Michalopoulos et al. 2011), efficient resources allocation (Duasa 2014; Sood and Ranjan 2015), and institutional efficiency (Okere et al. 2015), thus eventually promotes financial sector development.

A well developed and functioned financial sector is critically important for easy access to financial information with minimal costs, transaction costs reduction, fair investment decision, technological innovation, and growth stability. The technical innovation, according to Schumpeter (1911), critically important for economic growth but the effects of fiscal and financial innovation on economy receive little attention in empirical investigation. However, recent period financial innovation and its potential impact has attracted immense interest among researchers and encourages further investigation by considering the various aspect of the economy such as the economic growth (Qamruzzaman and Wei 2017, 2018b, 2018c; Bara et al. 2016; Bara and Mudxingiri 2016), on firms performance (Muthinja and Chipeta 2018; Carbó Valverde et al. 2016), on money demand (Dunne and Kasekende 2018; Kasekende 2016), on banking sector growth (Chipeta and Muthinja 2018; Kamau and Oluoch 2016), and many more. Financial innovation tends to accelerate the financial development allowing investment diversifications and risk minimization and thus plays a decisive role in economic growth (Bhatt and Mundial 1989). In addition, financial innovations augment the capital accumulation process in the financial system by encouraging savings propensity among the population with improved financial assists and intensify investment opportunity by offering innovative and less risky financial instruments. Most prominently, financial innovations open a gate for the undeserving

population in the society to come under the umbrella of the formal financial system and avail the benefit of finance.

In the finance literature, the contribution from financial innovation in the economy explained with three key aspects was observed. First, financial innovation expands economic activities by promoting financial inclusion, facilitating a financial transaction in international trade, enabling remittance, and uplifting financial efficiency. Second, the innovation-growth hypothesis postulated that financial innovation increases the quality of financial products and services (Schrieder and Heidhues 1995; McGuire and Conroy 2013), expedites the financial development process (Ozcan 2008), improves capital accumulation and allocation processes (Allen 2011), and increases the level of efficiency in financial institutions (Shaughnessy 2015). Third, financial innovation in the form of institutional development in the financial system expedites the financial process with greater accessibility to formal financial service, such as internet banking and mobile banking services (Raffaelli and Glynn 2013; Hargrave and Ven 2006), microfinance institutions, NGOs, and hybrid organizational forms (Battilana and Dorado 2010). The institutional availability with offering financial service improves the economy by including a greater number of people in the mainstream economic development process (Epstein 1992; Siddiqui and Ahmed 2009; Glaeser et al. 2004).

On the other hand, financial inclusion, in definition, is the ease of financial service access, availability, and the usage from formal financial institutions across the country. Innovative financial services, products, and financial institutions entice society to becoming habituated in using financial services from financial institutions, like the creation of accounts, borrowing funds, the use of ATMs, amongst others. Nonetheless, financial inclusion is the ultimate output with the adaption and diffusion of financial innovation. Therefore, the question can arise that do financial innovations promote the speed of financial inclusion in the financial system or in another way, do financial inclusion demands innovative financial instruments and services?

This study is novel in various aspects. First, with the study, for the first time, the financial innovation index was developed as a proxy of financial innovation rather relying on a single indicator. Even though the existing empirical literature had shown that a number of proxy indicators were used to address financial innovation in the equation, no consensus indicators appear in this regard. Therefore, this study tried to mitigate this gap by considering the financial innovation index with three (03) proxies, which have been repetitively used in different studies. Second, though empirical literature produces evidence regarding the financial inclusion index measuring the financial inclusion effects no such study had been performed yet nevertheless. Third, so far, to the best of the authors knowledge, this is the first ever-empirical investigation focusing on the nexus between financial innovation and financial inclusion.

The remaining structures of the article are as follows. Detailed empirical literature allied to present research in Section 2 is discussed. Section 3 deals with the research variables definition along with the details of the different econometrical methodologies used in empirical investigation. The mode estimation and its interpretation exhibited in Section 4, and finally, the summary findings and policy implications are explained in Section 5.

## 2. Literature Reviews

The nexus between financial innovation and financial inclusion has yet to be tested, nonetheless, a vast number of researchers have shown their keen interest in exploring the effects running from financial inclusion and financial innovation to different aspects of the economy. With this connection, pertinent study findings were summarized tagging with either financial innovation or/and financial inclusion.

### A. Financial Innovation and Its Role Understanding from the Empirical Literature

Financial innovation, in the Miller (1986) view, has been a critical and persistent ingredient for economic progress because of the financial markets with financial innovation are able to produce a multitude financial instruments, alternative risk transfer assets, and variants tax-deductible equity. Although, the importance of financial innovation in the modern financial system is well

acknowledged and receives minimal attention from financial experts, researchers, policymakers, and development agency.

However, a group of researchers put their considerable efforts of establishing the nexus financial innovation-led economic growths and produced substantial evidence in favor of a positive association between economic growth and financial innovation see, for example (Qamruzzaman and Wei 2017, 2018a, 2018c; Laeven et al. 2014, 2015; Michalopoulos et al. 2009, 2011; Bara and Mudxingiri 2016; Bara et al. 2016). They argue that financial innovation expands economic activities through capital accumulation, efficient financial intermediation, and financial institutional development. Besides that, financial innovation is also dealing with financial instruments development, corporate structure, financial reporting and techniques, and overall financial sector development.

Explaining the financial innovation-growth nexus, in accordance with existing empirical findings, four types of the causal hypothesis available were observed. First, the supply leading hypothesis that is, financial innovation promotes economic growth by allowing financing expansion, trade efficiency, easy access to financial services, and efficiency in financial institutions of dealing with a customer (Beck 2010; Shittu 2012). Second, the demand-leading hypothesis that is, economic growth expands economic activities in both macro and micro level. Therefore, financial services availability is imperative to maintain the normal speed of aggregated economic progression. Third, the feedback hypothesis that is caused by both financial innovation and economic growth is also known as bidirectional causality. The feedback hypothesis explained that the effect could be observed from each other and empirical literatures have produced ample evidence in this regards see, (Bara and Mudxingiri 2016; Bara et al. 2016; Qamruzzaman and Wei 2017, 2018a, 2018b, 2018c). Fourth, the neutral hypothesis implies that no causality exists between financial innovation and economic growth. In their respective studies, Lumpkin (2010) and Sekhar and Gudimetla (2013) found evidence confirming no causality between financial innovation and economic growth.

Financial innovation, according to Bhatt and Mundial (1989), reduces the risk and transaction costs in the financial system through effective and efficient payment mechanisms, institutional efficiency and thus accelerates capital market development. Financial innovation plays both objective and subjective roles in financial development, such as increased savings propensity in the society by offering innovative financial assets and the accumulation of capital for investment to increase output. Financial innovation in the financial system leads to financial diversity by introducing diversified financial instruments each of them possess unique the attributes and features. These diversifications in financial assets and services encourage savings propensity in the society in the form of financial assets and borrowing that ensure efficient allocation of economic resources in productive investment projects. Further, the efficient allocation of savings into productive investment augments financial activities and leads to ensure financial integration in the financial market, and thus allows financial development, at large.

The effects of financial innovation also discussed on operational performance in light of the efficiency of financial institutions, preferably bank-based financial institutions, such as Camelia and Angela (2011) investigated financial innovation and operational efficiency of Romanian banks spanning from 2002 to 2010. The data envelopment analysis was applied to reach conclusive evidence. The study findings unveiled foreign banks operating in Romania are more efficient than domestic banks. They postulated that foreign banks' efficiency rely on financial products and service diversifications and create customer-based operation. Further evidence relates to the financial innovation-led financial performance observed in the Chipeta and Muthinja (2018) study. In that study, they ascertain the positive association between financial innovation and operational performance in Kenyan banks based financial institutions. Similar findings relating to Kenyan banks performance with financial innovation was found in (Muthinja 2016; Makini 2010).

Financial innovation plays a critical role in the financial system in a two different way like product innovations, referring to the emergence of new and innovative financial instruments in the form

of financial assets and process innovation, referring to the efficient dispatch of financial services (Tufano 2003; Frame and Woosley 2004).

B.   **Financial Inclusion and Its Role Understanding from the Empirical Literature**

Growing empirical literature identified the effects of financial inclusion in the economy are versatile such as, augment consumption, productive investment, increase savings propensity, manpower empowerment (Ashraf et al. 2010; Dupas and Robinson 2009). Furthermore, access to financial service plays a critical role in reducing income inequality and poverty. A group of the researcher including, Mookerjee and Kalipioni (2010), Banerjee et al. (2018), Galor and Zeira (1993), and Beck et al. (2007) postulated in their respective study that a lack of access to financial services can augment income inequality and poverty in the economy. With a similar note, Swamy (2012) argued that the financial inclusion through the bank-based financial institutions accelerate access to finance to poor and positively influence the reduction of income inequalities in the economy and the financial intermediation boost inclusive economic growth.

A line of research findings available in finance literatures are those intended to explain the nexus Financial inclusion-led economic growth, see for example (Adeola and Evans 2017). In a study, Burgess and Pande (2005) documented the financial inclusion to foster economic growth through poverty alleviation. Similar findings were also experienced by a number for researchers in their studies including, Kim (2016) as observed in forty OECD countries, Babajide et al. (2015) as found in Nigeria, Sharma (2016) as spotted in the emerging Indian economy, and Kim et al. (2018) as unveiled for OIC countries. Financial inclusion extends the current consumption trend by allowing future investment opportunities, implying that easy access to financial services creates ample scope for fund accumulation by accepting financial assets, depositing money into the bank, availing credit facilities for investment, and diversifying the investment risk.

Second thoughts prevail in the empirical literature pertinent to financial inclusion that is the nexus between financial inclusion and financial development. Financial inclusion or banking sector outreach in the economy is the process of availing required financial service at a fair price, at the right place, and without any discrimination in the society. The prime target in financial inclusion should be beneficial to poor and undeserving people who are not using formal financial services. It implies that it thus brings the unbanked population into the formal financial system so that they are able to avail financial services such as savings, deposits, credit facilities, and insurance. The inclusive financial system entices savings propensity, capital accumulation, productive investment, and entrepreneurial development that assist in improving the standard of living in society (Demirgüç-Kunt and Klapper 2012). In addition, an inclusive financial system also reduces the possibility of emerging informal credit sources in the economy. Thus, the all-inclusive financial system ensures institutional efficiency, secure and safe savings and investments by facilitating all the range of efficient financial services. Therefore, sustainable financial development can be observed in the economy with effective and efficient implementation of financial inclusion.

Rasheed et al. (2016) investigated the role of financial inclusion on financial development spanning 2004–2012 in a panel of 97 countries with system-GMM estimation. They unveiled a positive association between financial inclusion and financial development. In a similar note, Allen et al. (2014) claimed that, in Africa, innovation in financial services, like mobile banking, has a positive effect on overcoming financial infrastructural limitations and allows the population to access financial services. The inclusion of the depriving and geographically located population in the mainstream of the financial system accelerates financial activities and simultaneously reduces the market fraction. Further evidence is found in the Adeola and Evans (2017) study. They investigated the relationship between financial inclusion, financial development and economic diversification in Nigeria by applying the fully modified OLS. The study findings disclosed that a significant effect on financial development from financial inclusion is proxy in terms of financial access and financial usages, respectively.

The reverse effects, implying financial development accelerates financial inclusion, also available in empirical studies. For example, Kumar (2013), explained in his study that banking institution

development allows greater access to formal financial services to the society, eventually increasing financial inclusion as a whole.

Apart from leading financial inclusion-led economic growth and financial development, observed a financial inclusion role was also observed in other economic aspects, such as, the reduction of income inequality (Mookerjee and Kalipioni 2010), the positive effects on foreign capital inflows (Qamruzzaman and Wei 2019), financial inclusion positively assisting in establishing financial stability, and poverty reduction (Yunus 2011; Chibba 2009).

*2.1. Motivation to Study Asymmetry Relationship*

The nexus between financial innovation and financial inclusions is yet to be unleashed through empirical investigation. Even though, empirical literature produced ample evidence focusing financial innovation with other macroeconomic variables, such as financial innovation-led economic growth, financial innovation-led financial development, and financial inclusion, such as financial inclusion-led financial development, financial inclusion-led financial development, and financial inclusion-led financial efficiency. Therefore, with the available nexus around financial innovation and financial inclusion, it can be presumed that there is a relationship between financial innovation and financial inclusion in the financial system.

The underlying motivation of investigating the asymmetric relationship between financial innovation and financial inclusion is to address the impact of positive and negative changes in financial innovation on financial inclusion and vice versa.

*2.2. Research Questions and Proposed Hypotheses*

The intended purpose of the study is not to unveil the key determinants for financial inclusion but rather to drag-out fresh insights through exploring the nexus between financial inclusion and financial innovation while incorporating two more variables namely, financial development and remittance inflows by applying pooled group mean (PGM) panel ARDL and panel nonlinear ARDL by following the proposed framework by Shin et al. (2014). Figure 1 depicts the summary of the proposed hypotheses, describing the direction of possible causality among these aforementioned variables. Pertinent to the current study, the following six (06) hypotheses were tested.

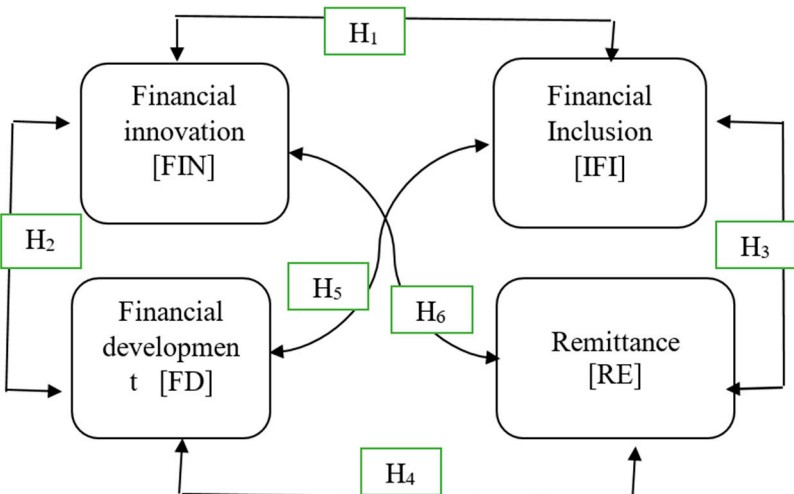

**Figure 1.** Conceptual framework of the possible pattern of causality between the variables.

$H_{A,B}^1$ Financial innovation Granger-cause financial inclusion and vice-versa

$H_{A,B}^1$ Financial innovation Granger-cause financial development and vice-versa

$H_{A,B}^1$ Financial inclusion Granger-cause remittance inflows and vice-versa

$H^1_{A,B}$ Financial development Granger-cause remittance inflows and vice-versa

$H^1_{A,B}$ Financial development Granger-cause financial inclusion and vice-versa

$H^1_{A,B}$ Financial innovation Granger-cause remittance inflows and vice-versa

## 3. Data and Methodology of the Study

This study explored the following: First, whether financial innovation positively induces the speed of financial inclusion. Second, the type of causality that is running between financial inclusion and financial innovation.

To do so, monthly cross-sectional data for six (06) countries representing the south Asian economy for the period 1990M1 to 2018M12 were collected. All the data used in this study collected from the central bank annual reports and online data archived of the respective countries see, Reserve Bank of India (2019); State Bank of Pakistan (2019); Central Bank of Sri Lanka (2019); Nepal Rastra Bank (2019); Royal Monetary Authority (2019) and Bangladesh Bank (2019).

For financial inclusion, empirical literature depicts the two lines of studies pertinent to financial inclusion proxy. One group of researchers has shown their keen interest to rely on a single proxy for financial inclusion representation in their empirical model. On the other hand, another group of researchers devoted to constructing the financial inclusion index by taking a number of proxy indicators with the construct procedure developed by Sarma (2008). This study followed the second line of thought, that is, the construction of a financial inclusion index rather than relying on a single proxy indicator. Therefore, a financial inclusion index was constructed with the application of the indexing procedure initiated by Sarma (2008) (see Appendix A for more details discussion relating to financial inclusion index construction).

For financial innovation, the selection of a single indicator for capturing the effect of financial innovation in an empirical model was not wise because in the empirical literature, the authors observed a number of proxy indicators that were used by researches in their studies. Therefore, for the first time, we developed a financial innovation index with three (03) indicators was developed for wide use in different studies. (For details of the variable definition and the index construction, please see Appendix A). The principal component analysis techniques were applied for the financial innovation index construction.

Apart from financial innovation and financial inclusion, two macroeconomic fundamentals were also considered namely, financial development and remittance inflows in the economy as control variables in the equation. From the motivation of incorporating those two variables that are in the empirical literature, this study observed that both financial development and remittance play a directive role in the financial system, therefore, acknowledging the possible effects of financial development and remittance on financial innovation and financial inclusion also addressed in empirical estimation. All variables are presented in logarithmic form. The descriptive statistics of research variables are presented in Table 1.

**Table 1.** Summary of descriptive statistics.

| Description | Obs | Mean | Stdard |
|---|---|---|---|
| Depositors with commercial Banks | 4032 | 35.9161 | 11.3339 |
| ATMs per 100,000 adults | 4032 | 92.1450 | 23.5900 |
| Commercial bank branches per 100,000 adults | 4032 | 8.0251 | 6.0510 |
| Credit to the private sector | 4032 | 69.2246 | 25.0111 |
| The ratio of aggregate money supply to narrow money | 4032 | 3.5131 | 0.3608 |
| The ratio of Broad to narrow money | 4032 | 4.0108 | 0.2058 |
| The percentage change in domestic credit to the private sector | 4032 | 0.0040 | 0.0443 |
| Domestic credit to private sector (% of GDP) | 4032 | 10.7436 | 3.6621 |
| Per capita remittance received | 4032 | 4.4871 | 2.6934 |

### 3.1. Modeling and Methodological Framework

The objective of this study is to explore new insights by explaining the nexus between financial inclusion and financial innovation along with two control variables namely, financial development and remittance inflows in South Asian countries. The generalized empirical model can be represented in the following ways:

$$IFI_{it} = \beta_0 + \beta_{1t}FD_{it} + \beta_{1t}RE_{it} + \beta_{1t}FIN_{it} + \varepsilon_{it} \tag{1}$$

where *IFI* denotes financial inclusion, *FD* stands for financial development, *RE* represents remittance inflows, and *FIN* denotes for financial innovation. $\varepsilon_{it}$ for the residual term in the equation and assumed to be normally distributed.

Cross Sectional Dependency Test

The cross-sectional dependency test is imperative in panel data empirical investigation, in particularly, for representative countries containing similar economic attributes, like developing countries, emerging economies, and transition countries. Due to trade internationalization, financial integration, and globalization make a similar economy subject to experience the effect with any shock in other countries. Therefore, investigating the presence of cross-sectional dependency would most likely demand an empirical investigation with panel data. In the investigation, four tests have been widely used. The Lagrange multiplier (LM) test was proposed by Breusch and Pagan (1980), which is preferred in a situation when the cross-section (N) is smaller than time (T). Based on the following equation, the LM test statistics can be constructed:

$$y_{it} = \alpha_i + \beta_i x_{it} + u_{it} \ \ I = 1, \ldots N, \ t = 1, \ldots T \tag{2}$$

where $y_{it}$ denotes dependent variable, $x_{it}$ are the independent variables and the subscript of *t* and *I* represent for the cross-section and time period, respectively. The coefficients of $\alpha_i$ and $\beta_i$ respectively represent the country-specific intercept and slope in the equation. In the contest of the LM cross-section dependency test, the null hypothesis of cross-section independence—$H_O = \text{COV}(u_{it}u_{jt}) = 0$ for all *t*, and $t \neq j$, against the alternative hypothesis of cross-sectional dependence——$H_O = \text{COV}(u_{it}u_{jt}) \neq 0$ for at least $t \neq j$. Moreover, the LM test statistics can compute the following equation:

$$LM = T \sum_{i=1}^{N-1} \sum_{j=i+1}^{N} \hat{\rho}_{IJ} \xrightarrow{d} X^2 N(N+1)2 \tag{3}$$

where $\hat{\rho}_{ij}$ represents the pairwise correlation of the residuals.

The LM test is not suitable in a situation with a larger cross-section (*N*), therefore overcoming this limitation, Pesaran (2004) suggest the following: The Lagrange multiplier ($CD_{lm}$) that is the scaled version of LM test:

$$CD_{lm} = \sqrt{\frac{N}{N(N-1)}} \sum_{I=1}^{N-1} \sum_{J=i+1}^{N} \left(T\hat{\rho}_{ij} - 1\right) \tag{4}$$

Under a cross-sectional independence of the null hypothesis with $t \to \infty$ and then $N \to \infty$, $CD_{lm}$ test statistics follow an asymptotic normal distribution (see (Nazlioglu et al. 2011; Menyah et al. 2014; Wolde-Rufael 2014)). In the case of larger *N* relative to *T*, the $CD_{lm}$ estimation is subject to size dissertation. Therefore, Pesaran (2006) proposed the following CD test, which is suitable in a situation when *N* is larger than *T*:

$$CD_{lm} = \sqrt{\frac{2T}{N(N-1)}} \sum_{I=1}^{N-1} \sum_{J=i+1}^{N} \left(\hat{\rho}_{ij}\right) \tag{5}$$

The CD test followed an asymptotically standard normal distribution for investigating the null hypothesis of cross-sectional interdependency with $t \to \infty$ and then $N \to \infty$ in any order (Nazlioglu et al. 2011). Furthermore, the CD test might produce distorted information in a situation where the population average pairwise correlation is zero and the individual pairwise correlation is non zero. Limiting the negative effect, Pesaran et al. (2008) proposed the bias-adjusted LM test. $LM_{adj}$ utilize the exact mean and variance of the LM statisitcs in case of the large panel first $t \to \infty$ and then $N \to \infty$. The bias-adjusted LM statistics can be computed with the following equation:

$$CD_{lm} = \sqrt{\frac{2}{N(N-1)}} \sum_{I=1}^{N-1} \sum_{J=i+1}^{N} \left( \frac{(T-K)\hat{\rho}_{ij}^2 - u_{Tij}}{v_{Tij}^2} \right) \overrightarrow{d}(N,0) \tag{6}$$

where $k$ refers to the number of regresses, $u_{Tij}$ and $v_{Tij}^2$ specifies the mean and variance of $(T-K)\hat{\rho}_{ij}^2$, respectively.

### 3.2. The Symmetric Panel ARDL

The investigation begins with an assumption of the symmetric relationship between financial inclusion and financial innovation. Therefore, the framework study used is widely known as the pooled group mean (PGM) or panel ARDL estimation initially proposed by Pesaran and Smith (1995). Further development was performed by Pesaran et al. (1999) and a well-defined model was proposed to investigate the long-run association of dynamic panel data having variables integration in mix order, either I(0) or/and I(1).

Panel ARDL, according to Pesaran et al. (1999), possesses certain advantages concerning panel dynamic estimation such as fixed effects, random effects, instrumental estimation or the generalized method of moments (GMM) proposed by Anderson and Hsiao (1981), Arellano (1989), and Arellano and Bover (1995). These methods can produce spurious results unless the coefficients are identical across the countries (da Silva et al. 2018).

The basic assumptions of PGM are first, the error correction term is free from correlation biasness and the normally distributed regressors. Second, there is a long run relationship between the dependent and explanatory variable, and third, the long-run parameter remains the same across the countries. Pesaran proposed the following generalized form of Panel ARDL as an empirical structure:

$$y_{it} = \sum_{j=1}^{p} \beta_{ij} y_{i,t-j} + \sum_{j=0}^{q} \gamma_{ij} x_{i,t-j} + \mu_i + \epsilon_{it} \tag{7}$$

This study estimated both the mean group (MG) proposed by Pesaran and Smith (1995) and the pooled grouped mean (PGM) in order to ascertain the efficient estimator for empirical investigation. Based on the standard Hausman test, the estimate failed to reject the null hypothesis that is there is no difference between the mean group and pooled mean grouped estimation, implying that the pooled grouped mean estimation is preferable. Therefore, this study performs an empirical model estimation with pooled grouped mean proposed by Pesaran et al. (1999). The pooled grouped mean can efficiently perform notwithstanding the variable order of integration either I (0) or/and I (0) see, (Pesaran et al. 2001; Kim et al. 2010; Fang et al. 2015).

The generalized form of pooled group mean ARDL can be represented as follows

$$\Delta IFI_{it} = \beta_{0i} + \beta_{1t} IFI_{it-1} + \beta_{2t} FIN_{it-1} + \beta_{3t} FD_{it-1} + \beta_{4t} RE_{it-1} + \sum_{J=1}^{M-1} \gamma_{iJ} \Delta IFI_{it-J} +$$
$$\sum_{J=0}^{N-1} \gamma_{ij} \Delta FIN_{it-J} + \sum_{J=0}^{N-1} \gamma_{ij} \Delta FD_{it-J} + \sum_{J=0}^{N-1} \gamma_{ij} \Delta RE_{it-J} + \mu_i + \varepsilon_{it} \tag{8}$$
$$i = 1, \ldots, N;\ t = 1, \ldots, T$$

where the subscript $t$ is the number of periods and $i$ is the sample unit. The long-run coefficient can be found from $\beta_1 \ldots \beta_2$ and the short-run coefficient from $\gamma_{iJ} \ldots \gamma_{iJ}$. The long-run coefficients as computed

$-\frac{\beta_{2i}}{\beta_{1i}}; -\frac{\beta_{3i}}{\beta_{1i}}; \ and \ -\frac{\beta_{4i}}{\beta_{1i}}$ since in the long-run, it is assumed that $\Delta IFI_{it-J}$, $\Delta FIN_{it-J}$, $\Delta FD_{it-J}$, and $\Delta RE_{it-J}$ is equal to zero(0).

Equation (10) can re-specified to include an error correction term in the following ways:

$$\Delta IFI_{it} = \partial_i \rho_{it-1} + \sum_{J=1}^{M-1} \gamma_{iJ} \Delta IFI_{it-J} + \sum_{J=0}^{N-1} \gamma_{ij} \Delta FIN_{it-J} \sum_{J=0}^{N-1} \gamma_{ij} \Delta FD_{it-J} + \sum_{J=0}^{N-1} \gamma_{ij} \Delta RE_{it-J} + \varepsilon_{it} \qquad (9)$$

where $\rho_{i,t-1} = IFI_{it-1} - \varphi_{0i} - \varphi_{1i} FIN_{t-1} - \varphi_{2i} FD_{t-1} - \varphi_{3i} RE_{t-1}$ are the linear error correction term of each unit and the coefficient of $\partial_i$ is the speed of adjustment towards long-run equilibrium. The parameters $\varphi_{0i}$, $\varphi_{1i}$, $\varphi_{2i}$, and $\varphi_{3i}$ are computed as $\varphi_{0i} = -\frac{\beta_{0i}}{\beta_{1i}}$, $\varphi_{Ii} = -\frac{\beta_{2i}}{\beta_{1i}}$, $\varphi_{2i} = -\frac{\beta_{3i}}{\beta_{1i}}$ and $\varphi_{3i} = -\frac{\beta_{4i}}{\beta_{1i}}$ respectively. It is noticeable from both Equations (8) and (9) that there is a decomposition effect, i.e., positive and negative change.

### 3.3. Asymmetric Panel ARDL

Unlike symmetric relationship, the asymmetric investigation requires two additional sets of data representing positive shock and negative shocks in explanatory variables in the equation. This version of the empirical model known as non-linear panel ARDL allows for an asymmetric response from financial development, financial innovation, and remittance inflows to financial inclusion. In other words, under this scenario, the positive and negative shock from financial innovation, financial development, and remittance are not expected to have identical effects on financial inclusion. Thus, the asymmetric version of Equation (8) is represented as follows:

$$\Delta IFI_{it} = \beta_{0i} + \beta_{1t} IFI_{t-1} + \beta_{2i} FIN_{t-1} + \beta_{3i} FD_{t-1} + \beta_{4i} RE_{t-1}$$
$$+ \sum_{J=1}^{M-1} \gamma_{iJ} \Delta IFI_{it-J} + \sum_{J=0}^{N-1} \left( \gamma_{iJ}^+ \Delta FIN_{t-J}^+ + \gamma_{iJ}^- \Delta FIN_{t-J}^- \right) + \sum_{k=0}^{O-1} \gamma_{ik} \Delta FD_{t-k} + \sum_{r=0}^{P-1} \gamma_{ir} \Delta RE_{t-r} \qquad (10)$$
$$+ \mu_i + \varepsilon_{it}$$

$$\Delta FIN_{it} = \delta_{0i} + \delta_{1t} IFI_{t-1} + \delta_{2i} FIN_{t-1} + \delta_{3i} FD_{t-1} + \delta_{4i} RE_{t-1}$$
$$+ \sum_{J=1}^{M-1} \mu_{iJ} \Delta IFN_{it-J} + \sum_{J=0}^{N-1} \left( \mu_{iJ}^+ \Delta IFI_{t-J}^+ + \mu_{iJ}^- \Delta IFI_{t-J}^- \right) + \sum_{k=0}^{O-1} \mu_{ik} \Delta FD_{t-k} + \sum_{r=0}^{P-1} \mu_{ir} \Delta RE_{t-r} \qquad (11)$$
$$+ \pi_i + \varepsilon_{it}$$

where $FIN^+$ & $FIN^-$ stand for the positive and negative shock of financial innovation and $IFI^+$ & $IFI^-$ represent the positive and negative shock of financial inclusions. The long run coefficients are computed as $FIN^+ = \frac{-\beta_{2i}^+}{\beta_{1i}}$, $FIN^- = \frac{-\beta_{2i}^-}{\beta_{1i}}$. These shocks are computed as the positive and negative partial sum decomposition of financial innovation and financial inclusion in the following ways:

$$\begin{cases} FIN_i^+ = \sum_{k=1}^{t} \Delta FIN_{ik}^+ = \sum_{K=1}^{T} MAX(\Delta FIN_{ik}, 0) \\ FIN_i^- = \sum_{k=1}^{t} \Delta FIN_{ik}^- = \sum_{K=1}^{T} MIN(FIN, 0) \end{cases} \qquad (12)$$

$$\begin{cases} IFI_i^+ = \sum_{k=1}^{t} \Delta IFI_{ik}^+ = \sum_{K=1}^{T} MAX(\Delta IFI_{ik}, 0) \\ IFI_i^- = \sum_{k=1}^{t} \Delta IFI_{ik}^- = \sum_{K=1}^{T} MIN(IFI, 0) \end{cases} \qquad (13)$$

The error correction version of Equations (10) and (11) is as follows:

$$\Delta IFI_{it} = \tau_{1i} \xi_{it-1} + \sum_{J=1}^{M-1} \gamma_{iJ} \Delta IFI_{it-J} + \sum_{J=0}^{N-1} \left( \gamma_{iJ}^+ \Delta FIN_{t-J}^+ + \gamma_{iJ}^- \Delta FIN_{t-J}^- \right) + \sum_{k=0}^{O-1} \delta_{ik} \Delta FD_{t-k} + \sum_{r=0}^{P-1} \gamma_{ir} \Delta RE_{t-r} \qquad (14)$$
$$+ \mu_i + \varepsilon_{it}$$

The error correction term ($\xi_{it-1}$) captures the speed of adjustment to long-run equilibrium in panel asymmetric Equation (9). On the other hand, the associated coefficient explains how long it requires to reach in the long run equilibrium in the presence of shocks in an explanatory variable in the short run.

GMM-System Based Panel Granger-Causality Test

For specifying directional causality between financial inclusion, financial innovation, financial development, and remittance inflows, we followed the panel error correction model causality test discussed by Shabani and Shahnazi (2019) in their research work. Panel Granger—causality test with the system-GMM application perform with the two-phase. In the first, the long run model estimation with dynamic-OLS for retrieving the residuals. Second, the residual obtained from the DOLS estimation used as an error correction term with first lagged, which specified the existence of long-run causality in the model. The equations for the short run and long run causality estimation are presented below:

$$\Delta IFI_{it} = \beta_{1i} + \sum_{k=1}^{m} \beta_{11ik} IFI_{it-k} + \sum_{k=1}^{m} \beta_{12ik} FIN_{it-k} + \sum_{k=1}^{m} \beta_{13ik} FD_{it-k} + \sum_{k=1}^{m} \beta_{14ik} RE_{it-k} + \zeta_{1i} ECT_{it-1} + e_{1it} \tag{15}$$

$$\Delta FIN_{it} = \beta_{2i} + \sum_{k=1}^{m} \beta_{21ik} FIN_{it-k} + \sum_{k=1}^{m} \beta_{22ik} IFI_{it-k} + \sum_{k=1}^{m} \beta_{23ik} FD_{it-k} \\ + \sum_{k=1}^{m} \beta_{24ik} RE_{it-k} + \zeta_{3i} ECT_{it-1} + e_{2it} \tag{16}$$

$$\Delta FD_{it} = \beta_{3i} + \sum_{k=1}^{m} \beta_{31ik} FD_{it-k} + \sum_{k=1}^{m} \beta_{32ik} IFI_{it-k} + \sum_{k=1}^{m} \beta_{33ik} FIN_{it-k} + \sum_{k=1}^{m} \beta_{34ik} RE_{it-k} + \zeta_{3i} ECT_{it-1} + e_{3it} \tag{17}$$

$$\Delta RE_{it} = \beta_{4i} + \sum_{k=1}^{m} \beta_{41ik} RE_{it-k} + \sum_{k=1}^{m} \beta_{42ik} IFI_{it-k} + \sum_{k=1}^{m} \beta_{43ik} FIN_{it-k} + \sum_{k=1}^{m} \beta_{44ik} FD_{it-k} + \zeta_{3i} ECT_{it-1} + e_{4it} \tag{18}$$

where $p$ represents the optimal lag length, which is determined by using Akaike's information criterion (AIC), this study found optimal lag for the estimation is 2, ECT stands for error correction term for assessing long-run causality, and $e_{it}$ for error term.

The underlying principle of using the system-GMM in determining the causality test with the panel error correction is consistent and unbiasedness in estimation. It is because the OLS based estimation is biased and creates an endogeneity problem in estimation (Soto 2009; Combes and Ebeke 2011). Therefore, other econometric techniques are required.

The generalized method of moments (GMM) is a commonly used econometric methodology in panel data estimation with endogenous regressors. In the empirical literature, there are two types of GMM estimations used, the first difference GMM estimation proposed by Arellano and Bond (1991) and the system GMM estimation proposed by Arellano and Bover (1995), and further developments performed by Blundell and Bond (1998). The first difference GMM estimation suffers from week instrument and a small sample size when endogenous variables are close to a random walk (Blundell and Bond 1998). The emergences of system-GMM estimation overcome the weaknesses in first difference GMM estimation (Arellano 2003; Baltagi 2008; Baum et al. 2007; Han et al. 2014). The system-GMM preforms estimating in two system equations. First, the original levels equation with a suitable lagged first difference as instruments and first difference equation with suitable lagged level as instruments. The application of system-GMM reduces the finite sample biased and increased the consistency in estimation (Blundell and Bond 1998). Therefore, system-GMM estimation was performed by using prior developed Equations (15)–(18).

The short-run and long run causality, after system GMM estimation, is identified by applying a standard Wald test. The null hypothesis of no causality is rejected if the coefficients of $\beta_{11}$ to $\beta_{44} = 0$ and the coefficient of ECT is statistically significant to ascertain the existence of long run causality in the equation.

## 4. Results and Discussion

### 4.1. Panel Unit Root. Cross-Section Dependency Test, and Cointegration Test

To test stationarity in the data set, several unit root tests were performed in accordance with existing empirical literates. The panel unit root test includes namely, Levin, Lin and Chu t proposed by Levin et al. (2002), the Breitung t-stat proposed by Breitung (2001), Im, Pesaran and Shin W-stat proposed by Im et al. (2003), and ADF-Fisher Chi-square proposed by Maddala and Wu (1999) test, having the null hypothesis that all the panels contain a unit root test and the Hadri Z-stat proposed by Hadri (2000) with the null hypothesis that all the panels are stationary. Table 2 exhibits the results of unit root tests. The study findings exposed that both variables were stationary after the 1st difference I(1) in all tests except the Hadri-z test, which confirms the variables were stationary at a level I(0).

**Table 2.** Panel unit root test.

| | Test | | | | | | | |
|---|---|---|---|---|---|---|---|---|
| | At Level | | | | After 1st Difference | | | |
| **Method** | **IFI** | **FI** | **FD** | **RE** | **ΔIFI** | **ΔFI** | **ΔFD** | **ΔRE** |
| Null: unit root (assumes common unit root process) | | | | | | | | |
| LLC—t (Levin et al. 2002) | −1.993 | 1.228 | 0.258 | 0.270 | −4.715 *** | −4.458 *** | −1.944 ** | −3.475 *** |
| Breitung t-stat (Breitung 2001) | 1.132 | 0.448 | 1.963 | 1.159 | −1.471 *** | −3.083 *** | −2.375 *** | −3.730 *** |
| Null: Unit root (assumes individual unit root process) | | | | | | | | |
| IPS W-stat (Im et al. 2003) | 0.903 | 1.995 | 1.692 | 0.891 | −2.876 *** | −2.959 *** | −5.339 ** | −1.877 *** |
| ADF—Fisher Chi-square (Maddala and Wu 1999) | 5.535 | 1.307 | 1.555 | 4.153 | 22.839 *** | 23.877 *** | 18.156 ** | 17.194 ** |
| PP—Fisher Chi-square | 10.187 | 0.8541 | 1.060 | 11.825 | 24.090 *** | 20.996 *** | 38.73 *** | 53.313 *** |
| Null hypothesis: no unit root with the common unit root process | | | | | | | | |
| Hadri Z-stat (Hadri 2000) | 4.489 | | 4.372 *** | 5.407 | 0.422 | | | 1.776 |

Note: ***, ** indicates level of significance at a 1% and 5%, respectively.

Table 3 exhibits the results of a cross-section dependency test. It was observed that the associated *p*-value of all four models' output is statistically significant at 1% of the level of significance. Thus, this confirms the rejection of the null hypothesis and in another way, the presence of cross-section dependence in the researcher variable can be assumed. Therefore, one can expect common dynamisms available in financial inclusion, financial innovation, financial development, and remittance inflows.

**Table 3.** Cross section dependency test.

| **Test** | **IFI/FIN, FD, RE** |
|---|---|
| $LM_{BP}$ (Breusch and Pagan 1980) | 50.527 (0.000) |
| $LM_{PS}$ (Pesaran 2004) | 12.854 (0.000) |
| $CD_{PS}$ (Pesaran 2006) | 6.896 (0.000) |
| $LM_{adj}$ (Pesaran et al. 2008) | 12.700 (0.000) |

In the next, the model estimation involves assessing the possible existence of cointegration between financial innovation and financial development by applying a panel cointegration test suggested by Pedroni (1999, 2004); and Kao (1999). Table 4 reports the results of the panel cointegration test. The panel cointegration test by model specification by Pedroni (1999, 2004) produced 11 test statistics based on the within-dimension and between-dimension. It is apparent that eight (08) out of eleven (11) test statistics are statistically significant at a 1% level of significance. These findings conclusively rejected the null hypothesis no cointegration by confirming the long-run association between financial innovation, financial inclusion, remittance, and financial development in South Asian countries. Further, the long-run association between financial innovation, financial inclusion,

financial development, and remittance inflows was also established in Kao (1999) panel cointegration model specification test.

**Table 4.** Panel cointegration test.

| Alternative Hypothesis: Common AR Coefficients (within-Dimension) | | |
|---|---|---|
| | **Statistic** | **Weighted Statistic** |
| Panel v-Statistic | 12.3317 *** | 7.5106 *** |
| Panel rho-Statistic | 12.4229 *** | 13.4849 *** |
| Panel PP-Statistic | 0.7521 | 0.5832 |
| Panel ADF-Statistic | −1.6157 *** | −1.6267 *** |
| **Alternative Hypothesis: Individual AR Coefficients (between-Dimension)** | | |
| | **Statistic** | |
| Group rho-Statistic | 1.9559 | |
| Group PP-Statistic | −3.8897 ** | |
| Group ADF-Statistic | −1.4324 *** | |
| Kao (1999): Cointegration test | | t-Statistic |
| ADF | | −0.5152 *** |

Note: ***, ** indicates level of significance at a 1% and 5% respectively.

In the next step, this study moves further towards the cointegration test with Westerlund (2007), because this test can be performed efficiently in either case that is existent and nonexistent of cross-sectional dependence. Westerlund proposed cointegration test produces four test statistics, two for Group and two for Panel ($G_t, G_\alpha$ and $P_t, P_\alpha$ ), of testing the null hypothesis that is no cointegration. Table 5 reports the results of the Westerlund Panel cointegration test. Considering the test statistics and associated $p$-value, it is convincing to reject the null hypothesis of no cointegration. That means, in the long run, all the variables move together regardless of their common dynamic association.

**Table 5.** Westerlund Panel cointegration test.

| Test Statistics | Value | *p*-Value |
|---|---|---|
| $Group_t$ | 4.719 | 0.000 *** |
| $Group_a$ | 2.939 | 0.009 *** |
| $Panel_t$ | 9.055 | 0.014 ** |
| $Panel_a$ | 12.005 | 0.000 *** |

Note: ***, ** indicates statistically significant at a 1% and 5% level of significance.

*4.2. Empirical Model Estimation without Asymmetry*

In the next step, the model estimation involves Panel ARDL (using Equations (10) and (11)) of identifying the coefficients elasticity both in the long run and short run. Table 6 exhibits the results of model estimation without asymmetry, where the results with financial inclusion as the dependent variable reported in column [1] and column [2] depict the results with financial innovation as a dependent variable in the equation, respectively.

For the long run, referring to the output reports in column [1] with financial inclusion as a dependent variable, the study findings unveiled the long-term positive influence running from financial innovation to financial inclusion that is the coefficient of financial innovation (a coefficient of 0.771 ***) is positive and statistically significant at a 1% level of significance. In particular, a 10% increase in financial innovation results in 7.71% development in financial inclusion. This finding suggests that further development in financial innovation that is emergence, adaptation, and diffusion of innovative financial assets, services, and instruments in the financial system can produce a positive progress in financial inclusion. The possible development in financial inclusion can be observed

with financial innovation embraced in the financial system in South Asian countries. It is because financial diversifications, the expansion of financial services coverage and offering improved financial instruments in the financial system, which is the ultimate result from financial innovation thus, assists in bringing financially deprived population into the formal financial system.

**Table 6.** Model estimation results without asymmetry.

| | Empirical Model Estimation | |
| --- | --- | --- |
| | **Financial Inclusion as Dependent Variable [1]** | **Financial Inclusion as Dependent Variable [2]** |
| Long-run elasticities | | |
| FIN | 0.771 *** (0.306) | - |
| IFI | - | 0.566 *** (0.167) |
| FD | 0.010 *** (0.003) | 0.776 * (0.017) |
| RE | 0.032 *** (0.008) | 0.108 ** (0.082) |
| Short-run elasticizes | | |
| ECT(−1) | −0.859 *** (0.093) | −0.582 ** (0.576) |
| ΔFIN | 0.254 *** (0.490) | |
| ΔIFI | | 0.095 ** (0.824) |
| ΔFD | 0.048 ** (0.010) | 0.027 (0.055) |
| ΔRE | −0.043 (0.022) | 0.042 **(0.108) |
| C | 0.161 *** (0.351) | −1.80 ** (1.651) |
| Hausman test | 1.02 (0.627) | 0.92 (0.342) |
| Log-likelihood | 100.6903 | 83.81886 |

Note: ***, ** indicates level of significance at a 1% and 5% respectively.

In regards to controlling variables that are financial development and remittance inflows, the effect running to financial inclusion also observed positively linked. More precisely, the effect of financial development (a coefficient of 0.010) being positive in sign and statistically significant at a 1% level of significance, is implying that future financial development in the south Asian economy can boost the speed of the financial inclusion process in the economy. The underlying reason for augmentation in financial inclusion is financial services availability, institutional effectiveness, and services efficiency, which play a motivational role in encouraging people to enjoy existing financial facilities, and eventually large segments of the population will be under the head of the financial system. On the other hand, foreign remittance inflows induce (a coefficient of 0.032) positive progress in financial inclusion. This study observed, in particular, 10% additional inflows of foreign remittance in the economy can result in the acceleration in the speed of financial inclusion by 0.32%. This finding suggests that excess money flows to households encourage them to transform their status from unbanked to financial integration by availing financial instruments for savings with the intention of future financial security and financial services for executing financial transactions, such as mobile banking, internet banking, and so forth.

For the short-run, the coefficient of the lagged error correction term (a coefficient of −0.895) is negative and statistically significant at a 1% level of significance, which is confirming the average speed of correction towards the long run equilibrium from any prior year shocks is considerable. Dealing with short-run elasticities running from financial innovation, financial development, and remittance inflows, this study observed both the financial innovation (a coefficient of 0.254) and financial development (a coefficient of 0.048) positively induced the process of financial inclusion. In particular, south Asian countries can experience positive development in financial inclusion by 2.54% and 0.48%, respectively with a 10% increase in financial innovation and financial development in the financial system. Meanwhile, foreign remittance produces a statistically insignificant impact on financial inclusion even though the elasticity of foreign remittance flow (a coefficient of −0.042) is negative in sign. The Hausman test ultimately shows that it is impossible to reject the homogenous constraint in long-term equilibrium at a 1% level of significance, meaning the PMG estimator is suitable and effective for estimation of the pooling long-term coefficients.

Referring to the results reported with financial innovation as the dependent variable (see Table 7, column [2]). For the long run, study findings disclosed the effect running from financial inclusion (a coefficient of 0.566), financial development (a coefficient of 0.776), and foreign remittance inflows (a coefficient of 0.108) in the development process of financial innovation in the financial system is positively linked and all the coefficients are statistically significant at a 1% level of significance. In particular, dealing with the financial inclusion effect on financial innovation, a 10% increase in financial inclusion was observed which can result in 5.66% development in financial innovation. The plausible interpretation is that with the increase of access to financial services by the population results in the higher financial demand in a diversified manner, implying that a financial system expects innovative financial assists, services and instruments availability for satisfying the continuous financial demand by the society. Thus, intensify financial innovation flourishment with the help of financial inclusion in the economy. Further, dealing with the nexus between financial development-led financial innovations, this study observed, in particular, 10% increases in financial development can augment the financial innovation process by 7.76%. This finding depicts financial development in the economy creates an ambiance favoring the embrace of new and innovative financial assets, services, and instruments by financial institutions so as to serve the growing financial demands in the economy. The study also divulges the positive association between foreign remittances inflows and financial innovation that is the 10% additional inflow of remittance results 1.08% enhancement in financial innovation. The possible interpretation stands in explaining the relationship that is households having an excess money supply, which induces savings propensity with future investment. Therefore, financial system experienced investment diversification demand from households and induced financial institutions to adopt innovative financial assets and services for satisfying the persistent demand from potential investors.

**Table 7.** Empirical model estimation with asymmetry.

| Model Estimation | | |
| --- | --- | --- |
| **Financial Inclusion as Dependent Variable [1]** | | **Financial Innovation as Dependent Variable [2]** |
| Panel—A: Long-run model coefficients | | |
| $FIV^+$ | 0.260 *** (0.187) | |
| $FIV^-$ | 0.705 ** (0.548) | |
| $FIC^+$ | | 0.036 ** (0.098) |
| $FIC^-$ | | 0.443 *** (0.355) |
| FD | 0.025 *** (0.078) | 0.218 *** (0.109) |
| RE | 0.031 ** (0.090) | 0.115 *** (0.069) |
| Panel—B: Short—rum model coefficients | | |
| ECT(−1) | −0.345 ** (0.329) | −0.532 *** (0.217) |
| ΔFIV+ | 0.987 *** (0.4267) | |
| ΔFIV- | 0.752 *** (0.443) | |
| ΔIFI+ | | −0.478 (0.201) |
| ΔIFI- | | 0.478 *** (0.136) |
| ΔFD | 0.197 ** (0.157) | 0.160 * (0.195) |
| ΔRE | 0.023 (0.019) | 0.403 *** (0.128) |
| Panel—C: Test of symmetry | | |
| $W_{LR}$ | 6.973 *** | 15.220 *** |
| $W_{SR}$ | 11.983 *** | 15.342 *** |
| Hausman test | 11.542 (0.416) | 9.348 (0.4994) |
| Log-likelihood | 128.394 | 273.983 |

Note: ***, ** indicates statistically significant at a 1% and 5% level of significance, respectively. Values in () are standard error.

For the short run, the error correction term (a coefficient of −0.582) is negative and statistically significant at a 1% level of significance, implying the existence of the short-run association. Considering the coefficients elasticity, it is obvious that financial inclusion (a coefficient of 0.095), financial development (a coefficient of 0.027), and remittance inflows (a coefficient of 0.042) are positively supplementing the process of financial innovation development. However, only the impact running from financial inclusion and remittance inflows are statistically significant at a 1% level of significance. The findings suggest that a growing trend in financial innovation in the short-run can be observed with further improvement in financial inclusion and foreign remittance inflows in South Asian countries. The Hausman test to specify model construction and validation, produces a statistic of 0.92 with a *p*-value of 0.342, providing evidence that PGM is consistent and more efficient in producing precise and reliable results with the pre-specified empirical model.

*4.3. Empirical Model Estimation with Asymmetry*

In this section, empirical model estimation involves the asymmetry that is the investigation of positive and negative shocks of the independent variables on the dependent variable. The empirical model estimation results exhibited in Table 7, column [1] contains model estimation with financial inclusion as a dependent variable and column [2] contains model output with financial innovation as a dependent variable, respectively.

Considering the results presented in column [1], assessing the long run and short-run asymmetry effects of financial innovation, a standard Wald test was executed with the null hypothesis that is "there are symmetric effects running from financial innovation to financial inclusion. The *Wald test* F-statistics for the long run (a coefficient of $W_{LR} = 6.973$) and for the short-run (a coefficient of $W_{SR} = 11.983$) clearly reject the null hypothesis in both situations. Alternatively, the *Wald test* statistics confirm the asymmetry effect running from financial innovation to financial inclusion. In regards to model consistency and precision in the empirical estimation, the Hausman test produces statistics of 11.542 with a statistically insignificant *p*-value of 0.416, confirming that the empirical model is consistent and more efficient in producing variables elasticities.

For the long run, it is ostensible that both positive shocks (a coefficient of 0.260) and negative shocks (a coefficient of 0.750) in financial innovation are positively linked with financial inclusion. These study findings imply that future changes experienced by financial innovation in either direction that increases or decrease in both situations, financial inclusion is affected. In particular, a 10% increase in positive shocks in financial innovation can result in 2.6% growth in financial inclusion, meanwhile, a 10% decrease in financial innovation can cause 7.05% declined trend in financial inclusion. The magnitude of the negative shock is greater than the elasticity of the positive shock of financial innovation on financial inclusion. On the other hand, for the short-run it is palpable that, like long run model, the positive shocks (a coefficient of 0.987) and negative shocks (a coefficient of 0.752) in financial innovation are positively linked to financial inclusion. More precisely, a 10% growth in positive shocks in financial inclusion can result in a 9.87% improvement in financial inclusion, meanwhile, a 10% decrease in financial inclusion can cause a 7.52% deterioration in financial inclusion. The study findings suggest that the contraction financial policy might have adverse causes in the process of financial innovation and the eventual outcome could be experienced by the economy by obstructing the development process of financial inclusion with greater intensity. Therefore, financial policy pertinent to financial stability and financial expansion is inevitable in order to augment the existing financial inclusion trend in the economy with innovative financial assets and services in the form of financial innovation.

For control variables, in the long run, the effect running from financial development (a coefficient of 0.025) and remittance inflows (a coefficient of 0.031) are positive and statistically significant at a 1% level of significance. Furthermore, in the short run, analogous to the long run, the influence running from financial development (a coefficient of 0.197) and remittance inflows (a coefficient of 0.023) also depicts positively with financial inclusion.

Referring to the results conveyed in column [2] with financial innovation as a dependent variable, the presence of asymmetry effects of financial inclusion on financial innovation both in the long run and in the short-run were investigated by applying the standard Wald test suggested with the null hypothesis of symmetry. The Wald test F-statistics in the long run (a coefficient of 15.220) and in the short-run (a coefficient of 15.342) are statistically significant at a 1% level of significance, implying the rejection of the null hypothesis. The study findings suggest asymmetry effects running from financial inclusion to financial innovation, which is applicable in both the long run and short run. In regards to the positive and negative shocks in financial inclusion, this study observed in the long-run, positive shocks (a coefficient of 0.036) and negative shocks (a coefficient of 0.443) were positively linked with financial innovation. In particular, a 10% increase in financial inclusion can result in a 0.36% increase in financial innovation. On the other hand, a 10% decrease in financial inclusion can decrease the financial innovation evolvement in the financial system by 4.443%. It is clearly manifested that negative shocks in financial inclusion produce significant magnitudes of positive shocks in financial inclusion. Therefore, it is imperative to formulate financial policies in such a way so as the existing process of financial inclusion moves with ease without facing any blockage, because of the impediment in financial inclusion adversely deterring the normal process of financial innovation.

For the short-run, the coefficient of the error correction term was observed to be negative and statistically significant, implying the existence of a short-run convergence between financial innovation and financial inclusion. Considering the short-run coefficients, the study findings divulge a positive shock in financial inclusion is (a coefficient of −0.478) negatively linked to financial innovation. However, the magnitude is statistically insignificant. Therefore, in the positive variation of financial inclusion did not make any considerable impact on financial innovation. In contrast, the negative shock (a coefficient of 0.478) in financial inclusion is also positively associated and the coefficient is statistically significant at a 1% level of significance. These findings suggest that a 10% decrease in financial inclusion results in a decrease trend in financial innovation by 4.78%. In line with the study findings regarding a negative variation in financial inclusion is critical for innovativeness in the financial system. That is why, the speed of financial inclusion might not be hampered due to financial policy, if so happened, financial innovation also affected at large.

For control variables, namely, financial development and remittance inflows, it is apparent from the finding that the coefficients elasticities, such as financial development (a coefficient of 0.443) and remittance inflows (a coefficient of 0.218), exhibited a positive relationship. The study findings suggest that future development in the financial sector could explore more opportunities in the economy for the adaptation and diffusion of financial innovation, which eventually allows greater financial diversity with innovative financial instruments, services, and institutions. Further, the continual inflows of foreign remittance also intensifies the demand for financial products and services that are the processes of financial innovation will receive a positive injection for further development.

### 4.4. Post Model Estimation with System-GMM Specification

In the next step, this study moved to investigate the robustness of the pre-specified empirical model for explaining the nexus between financial inclusion and financial innovation. Table 8 reports the results of the System-GMM empirical model. Panel-A indicates the short-run model estimation with a symmetry test along with the coefficient of error correction term and Panel-B exhibits the long-run model estimation with the symmetry test. From the model stability and validity diagnostic test statistics that are the conventional AR (2) and Sargan test, it was observed that the null hypothesis was not rejected at a 1% level of significance, implying that all regressors were valid instruments. This conclusion is applicable for both models. For investigating the symmetric relation, a standard Wald test was performed with the null hypothesis of symmetry in both the long run and short run. For the short run, the findings from the Wald test with financial inclusion as the dependent variable was (a coefficient of $W_{SR} = 12.705$) and Wald test with financial innovation as the dependent variable

was (a coefficient of $W_{SR}$ = 18.873). This finding suggests that in the short-run, the relationship between financial innovation and financial inclusion is asymmetric.

**Table 8.** Short- and long-run generalized method of moments (GMM) estimates and symmetry tests.

| | Model Estimation | |
|---|---|---|
| | **Financial Inclusion as Dependent Variable [1]** | **Financial Innovation as Dependent Variable [2]** |
| | Panel-A: short-run coefficients | |
| IFI(−1) | 0.537 *** [0.108] | |
| FIN(−1) | - | 0.857 *** [0.060] |
| ΔIFI+ | - | 0.029 ** [0.659] |
| ΔIFI− | - | 0.815 ** [0.620] |
| ΔFIN+ | 0.059 ** [0.064] | - |
| ΔFIN− | 0.026 ** [0.036] | - |
| Speed of adjustment | 0.534 | 0.763 |
| AR(2) | 0.418 (0.675) | −1.324 (0.553) |
| Sargon test | 55.694 (0.156) | 63.260 (0.723) |
| $W_{SR}$ | 12.705 ** | 18.873 (0.000) |
| | Panel-B: long-run | |
| IFI+ | - | 0.560 ** [0.252] |
| IFI− | - | 0.255 *** [1.318] |
| FIN+ | 0.064 *** [0.035] | - |
| FIN− | 0.033 *** [0.088] | - |
| $W_{LR}$ | 21.607 *** | 25.983 *** |
| | Control variable | |
| FD | 0.016 *** [0.074] | 1.118 *** [0.148] |
| RE | 0.018 *** [0.237] | 0.769 *** [0.102] |

None: **, *** denotes level of significant at a 5% and 1% respectively. Values at parenthesis () indicates standard error.

Finally, in the long-run symmetry, this study observed that the Wald test statistics (a coefficient of $W_{LR}$ = 20.607, and $W_{LR}$ = 25.983) in both models were statistically significant at a 1% level of significance. The results of the Wald test ascertain the existence of an asymmetric relationship between financial innovation and financial inclusion. This conclusion is applicable for both empirical models.

### 4.5. Panel Granger-Casualty with System-GMM Specification

This section moved to investigate the directional causality between financial inclusion, financial innovation, financial development, and remittance inflows. To accomplish this, the process initiated by Shabani and Shahnazi (2019) with the system-GMM specification was followed. The results of the causality test are exhibited in Table 9. The presence of long-term causality in the empirical model can be ascertained by observing the coefficients of ECT of each model. For long-run causality, the coefficient of error correction term, in particular when financial inclusion, financial innovation, and remittance inflows are considered as the dependent variable, are negative in sign and statistically significant at a 1% level of significance. More precisely, the study findings divulge bidirectional casualty between financial innovation and financial inclusion [IFI ↔ FIN], implying that in south Asian economy feedback hypothesis holds in explaining the causal relationship between financial inclusion and financial development.

For short-run causality, the study findings unveiled a number of the casual relationship among research variables. In particular, the feedback hypothesis holds for explaining the relationship between financial inclusion and financial innovation [IFI ↔ FIN], remittance inflows and financial inclusion [IFI ↔ RE]. These findings suggest that in the short run, any further development in either variable namely, financial inclusion, financial innovation, and remittance inflows, show that the ultimate effects can be observed in the associated variables. Furthermore, the study findings unveiled unidirectional causality running from financial development to financial inclusion [FD → IFI] and financial development to remittance inflows [FD → RE].

**Table 9.** Causality test with GMM specification.

| Dependent Variable | Short-Run Casualty | | | | Long-Run Causality | |
|---|---|---|---|---|---|---|
| | IFI | FIN | FD | RE | ECT(−1) | Remarks |
| IFI | - | 8.347 *** | 0.554 | 14.682 *** | −0.0253 *** | Presence |
| FIN | 12.250 *** | - | 4.534 ** | 1.411 | −0.165 ** | Presence |
| FD | 0.155 | 13.092 *** | - | 1.854 | 0.135 | |
| RE | 12.180 *** | 1.774 | 9.184 *** | | −0.773 *** | Presence |

Note: ***, ** indicates statistically significant at a 1% and 5% level of significance, respectively.

## 5. Summary and Concluding Remarks

The relationship between financial innovation and financial inclusion is yet to undergo extensive empirical investigation. This study, therefore, intended to mitigate the existing research gap by unsheathing new insights pertinent to explain how financial inclusion and financial innovation behave in the financial system due changes appeared in either variable. As a sample, we considered six (06) South Asian countries covering a span of period 1990M1−2018M12. This study used monthly data for empirical model estimation, which was exported from a central bank database of respective countries. For the empirical investigation, a number of econometric methodologies were employed including, PGM Panel ARDL proposed by Pesaran et al. (1999) and non-linear Panel ARDL by the following model specifications proposed by Shin et al. (2014). Furthermore, by establishing the directional causality between financial innovation, financial inclusion, remittance, and financial development, the Granger-causality test with System-GMM specification following Shabani and Shahnazi (2019) were applied. The key findings of this study are stated below:

First, referring to empirical model estimation with symmetry assumption (see, Table 7). Study findings established a positive association between financial innovation and financial inclusion both in the long run and short run. These findings suggesting that financial sector development either encouraging adaption of improving financial services, instruments and institutions can progressively encourage financial innovativeness in the financial system and vice-versa. Furthermore, for ensuring financial efficiency or ensuring financial services easy access to the society and the pull unbanked population into the formal financial system in both cases, the eventual effects can be observed in financial inclusion and financial innovation. Hence, it can be assumed that a bidirectional relationship prevails between them.

Second, the empirical model estimation with asymmetry assumption, study findings suggesting that both financial innovation and financial inclusion will be experienced greater intensity in either case of improvement in the financial system such as financial innovativeness or financial integration. Considering the asymmetric response that is positive and negative shocks, we observed that in the case of financial innovation both positive and negative shocks are positively linked with financial inclusion. On the other hand, the asymmetric effects of financial inclusion that is positive and negative shocks on financial innovation also positively associated. Therefore, it is important to monitor and take necessary initiatives by financial regulatory authorizes concentrating financial development so that the growth trend in financial innovation and financial inclusion remain stable, especially in the long term.

Third, considering the output of the Granger causality test. A feedback hypothesis holds explaining the causality between financial inclusion and financial innovation both in the long run and short run. The study findings suggested that financial sector development with either variable amplification that is, financial innovation or financial inclusion, the subsequent effect be observed in another variable. On the other hand, any financial policies anticipated for limiting the financial activities results, not only adverse consequences in financial inclusion, but also in financial innovation.

Considering the results explain above, it is obvious that the intertwined relationship between financial innovation and financial inclusion is critically important for vibrant financial sectors. This is why government and policymakers should consider all aspects of financial innovation and financial inclusion affects, not only each other, but also impacts on economic activities so that fiscal policy can

effectively guide further development in financial innovation and financial inclusions. In addition, financial institutions should expand their financial activities by incorporating newly emerged financial assets and services that are effectively work-out in other countries and allow financial services to all with easy access.

**Author Contributions:** The concept and design of this article come from J.W. along with model formulation. Data collection, empirical study review of conceptual development and drafting done by M.Q. In final editing and overall development effort contribute by authors in the article, the ration of contribution equally likely.

**Funding:** This research received no external funding.

**Conflicts of Interest:** The authors declare no conflicts of interest.

## Appendix A Details of Financial Innovation and Financial Inclusion Index Construction
**Financial Innovation**

Financial innovation, according to Tufano (2002), is the process of emergence, diffusion, and popularization of new financial instruments, financial institutions, financial technologies, and financial markets in the economy. The presence of financial innovation in the financial system can be addressed in two different wings, such as product innovation and process innovation. Over the last decade, a number of proxy indicators were used in the empirical study of addressing the effects of financial innovation on various aspects. Aligning with existing literature in this study, we usd three proxy indicators (see, Table A1) were used and moved to developed the financial innovation index by applying principle component analysis (PCA), which is used as a proxy for financial innovation.

**Table A1.** Financial innovation proxy indicators.

| Indicator | Definition | Reference |
|---|---|---|
| M3/M1 | The ratio of Aggregate money supply to Narrow money | Dunne and Kasekende (2018); Hye (2009); Mannah-Blankson and Belnye (2004); Qamruzzaman and Wei (2017, 2018a, 2018b, 2018c) |
| M2/M1 | The ratio of Broad to narrow money | Qamruzzaman and Wei (2017, 2018a, 2018b, 2018c) |
| Growth of DCP | The percentage change in domestic credit to the private sector | Ajide (2015); Michalopoulos et al. (2011) |
| Financial innovation composite index | | |

*Financial Inclusion Index*

The importance of financial inclusion emerged due to nearly 3 billion of the population being excluded from formal financial services known as financial exclusion (world bank). The concept of financial inclusion is subject to country-specific financial exclusion and related macroeconomic variables. Therefore, no agreeable and consensus definition is yet to appear in finance literature. Financial inclusion, according to Gwalani and Parkhi (2014), is the way of availing and utilizing formal financial services at a lower cost and affordable to reduce informal accounts. In other words, financial inclusion means accessibility, availability, and the use of all formal financial services to all (Kumar and Mohanty 2011). It is implying that the provision of access to financial services with the minimum cost along with the efficient financial intermediation by the financial system.

Furthermore, addressing the effects of financial inclusion with single indicators is not so straightforward. Since over the period, researchers in empirical studies used a number of proxy indicators see, Table A2 and some of the study used the index by constructing more than one indicator.

**Table A2.** Financial inclusion proxy indicators.

| Dimension | Definition | Reference |
|---|---|---|
| Banking penetration | Depositors with commercial Banks | Adeola and Evans (2017); Evans (2015); Naceur et al. (2015); Sarma (2008, 2012); Mbutor and Uba (2013) |
| Access | ATMs per 100,000 adults | Adeola and Evans (2017); Mookerjee and Kalipioni (2010); Rasheed et al. (2016) |
| | commercial bank branches per 100,000 adults | Sarma (2008); Kumar (2013); Rasheed et al. (2016); Gimet and Lagoarde-Segot (2012) |
| Usage | Credit-credit to the private sector | Sarma (2008, 2012) |

Before empirically investigating the nexus between financial innovation and financial inclusion in south Asian countries, our own index of financial inclusion (IFI) was first constructed. Constructing the index of financial inclusion, the authors closely followed the multidimensional methodology proposed by Sarma (2008) with accessibility, availability, and usages of banking services.

First, availability has been measured by two proxy indicators namely, automated teller machines (ATM) per 100,000 adults and commercial bank branches per 100,000 adults following Adeola and Adeola and Evans (2017); Mookerjee and Kalipioni (2010); Rasheed et al. (2016); Sarma (2008); Kumar (2013); Rasheed et al. (2016); Gimet and Lagoarde-Segot (2012). Second, accessibility has been measured by the penetration of banking services proxied by the number of depositors with commercial banks per 1000 adults following Adeola and Evans (2017). Evans (2015); Naceur et al. (2015); Sarma (2008, 2012); Mbutor and Uba (2013). Third, the proxy was used to measure the usage dimension by the total deposits and credits relative to gross domestic product by following Sarma (2008, 2012). Table A2 represents the financial inclusion proxy indicators of south Asian countries in the year of 2017.

The study now proceeds to estimate the dimension index for each dimension by following the Sarma (2008) specification using the following formula:

$$d_i = \frac{A_i - m_i}{M_i - m_i} \tag{A1}$$

where $A_i$ is the actual value of dimension $i$, $M_i$ is the maximum value of dimension $i$, and $m_i$ is the minimum value of dimension $i$, respectively, the output from formula ensure that $0 < d_i < 1$. The higher the value $d_i$ indicates the higher the achievement in dimension $i$.

For availability dimension, first the dimension index was determined for each dimension, a weighted average dimension index construct by allowing a two-third weight for a bank branch and one-third for the ATM index for the availability dimension as suggested by Sarma (2008). The index of financial inclusion then measured, with a given weight such as 1 for the index of bank penetration, 0.5 for Availability and 0.5 for usage, by the normalized inverse Euclidean distance of the point Di from the ideal point $I = (1, 1, 1, \ldots 1)$. The exact formula is

$$IFI = 1 - \frac{\sqrt{(1 - P_i) + (0.5 - A_i) + (0.5 - U_i)}}{\sqrt{n}} \tag{A2}$$

This study also observed in empirical literatures that financial inclusion is also affected by other macroeconomic variables such as financial development, foreign remittance receipts, microfinance institutions, foreign direct investment, trade openness see, e.g., (Aga and Peria 2014; Aggarwal et al. 2011; Anzoategui et al. 2011; Chowdhury 2011). Therefore, enhancing estimation robustness, two more variables pertinent to the existing literature were included as the control variable namely, financial development and remittance received. To capture the effect of financial development in the model, the commonly used financial development indicator the ratio of broad money to GDP can

be considered see, (Calderón and Liu 2003; King and Levine 1993; Nyamongo et al. 2012), In addition, remittance inflows proxied by remittance inflows to GDP (%).

**Table A3.** Variables Descriptions and Sources.

| Variable | Indicators | Description | Data Sources |
|---|---|---|---|
| Financial inclusion | Banking penetration | Depositors with commercial Banks | Reserve Bank of India (2019); State Bank of Pakistan (2019); Central Bank of Sri Lanka (2019); Nepal Rastra Bank (2019); Royal Monetary Authority (2019) and Bangladesh Bank (2019). |
| | Access | ATMs per 100,000 adults | |
| | | commercial bank branches per 100,000 adults | |
| | Usage | Credit-credit to the private sector | |
| | Financial inclusion index | Authors calculation using multidimensional methodology proposed by Sarma (2008) | |
| Financial Innovation | M3/M1 | The ratio of Aggregate money supply to Narrow money | Reserve Bank of India (2019); State Bank of Pakistan (2019); Central Bank of Sri Lanka (2019); Nepal Rastra Bank (2019); Royal Monetary Authority (2019) and Bangladesh Bank (2019). |
| | M2/M1 | The ratio of Broad to narrow money | |
| | Growth of DCP | The percentage change in domestic credit to the private sector | |
| | Financial innovation index | Authors calculation using Principal component analysis (PCA) | |
| Financial development | Domestic credit to the private sector (% of GDP) | Domestic credit to private sector refers to financial resources provided to the private sector by financial corporations, such as through loans, purchases of nonequity securities, and trade. | International financial statistics (IMF) |
| Remittance Inflows | Per capital remittance received | Personal remittances comprise personal transfers and compensation of employees. Personal transfers consist of all current transfers in cash or in-kind made or received by resident | International financial statistics (IMF), World Development Indicator (WB) |

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
