# Peer review of "Financial Innovation and Financial Inclusion Nexus in South Asian Countries: Evidence from Symmetric and Asymmetric Panel Investigation"

_ijfs, doi:10.3390/ijfs7040061_

Round 1

Reviewer 1 Report

Major concerns

The authors need to reorganize the paper.

The authors also need to make significant editorial changes to the text and sentences in the manuscript because currently the paper is difficult to read due to the use of compound language and grammatical omissions and errors.

For instance, in section 1, (i) there shouldn’t be citation in the abstract; (ii) some sentences are difficult to read such as “Since study findings established a critical relationship then effective policy” and “Study findings with symmetry assumption,” (iii) the authors did not provide the contribution of the study to the existing literature. (iv) the definition of financial inclusion needs a citation in the second paragraph of the introductory section. (v) financial innovation was not defined in the introductory section.

In section 2, the conceptual framework is not well-explained, and for this reason, the hypotheses do not have any meaning to the reader. The authors can add a new section “hypothesis development” to develop the hypotheses better. Also, I think the authors only need two hypotheses not six (6) hypotheses - h1: financial inclusion affects financial innovation, and h2: financial innovation affects financial inclusion.

In empirical testing, no robustness test was conducted to validate the first hypothesis (?1?,?) which is most relevant to the research question

Author Response

Dear reviewer,

                Thank you so much for your meaningful and thoughtful comments and insightful suggestion for further development. During the revised period, we tried our best in incorporating your suggestions and recommendation so that we can uplift your expectation.

                Please, still if you feel we need to do more work for further improvement, we love to do that, let us know.

Thank you so much

Regards

Major concerns

The authors need to reorganize the paper. The authors also need to make significant editorial changes to the text and sentences in the manuscript because currently, the paper is difficult to read due to the use of compound language and grammatical omissions and errors.

Response: Dear reviewer, we are hoping that this time you may find the article in better shape since we work on your suggestions.

Reviewer response:

For instance, in section 1,

(i) There should not be citation in the abstract;

Response: rectified

(ii) some sentences are difficult to read such as “Since study findings established a critical relationship then effective policy” and “Study findings with symmetry assumption,”

Response: Dear reviewer, we rewrite and reconstruct for eliminating difficulties in reading

(iii) the authors did not provide the contribution of the study to the existing literature.

Response: Dear review, please see the third para on page 2. Where we pointed out the study novelty, which is the contribution of this study over existing literature.

(iv) the definition of financial inclusion needs a citation in the second paragraph of the introductory section.

Response: Dear reviewer, thank you so much, it was a mistake, now it is rectified

(v) financial innovation was not defined in the introductory section.

Response: Dear reviewer, thank you so much, by following your suggestion, now we insert the definition of financial innovation in the introductory section, please see the last para on page 2

In section 2, the conceptual framework is not well-explained, and for this reason, the hypotheses do not have any meaning to the reader. The authors can add a new section “hypothesis development” to develop the hypotheses better. Also, I think the authors only need two hypotheses not six (6) hypotheses - h1: financial inclusion affects financial innovation, and h2: financial innovation affects financial inclusion.

Response: Dear reviewer, thank you so much of your concern, actually this name of the section will be research questions and proposed hypothesis. Actually, we are focused on testing first two hypotheses; however, since we have control variables therefore, we test all six hypotheses in causality test to see their directional effects.

In empirical testing, no robustness test was conducted to validate the first hypothesis (?1?,) which is most relevant to the research question

Response: Dear reviewer, please see section 4.4 dealing with empirical model estimation with System-GMM for checking empirical model robustness.

Reviewer 2 Report

Referee Report

ijfs-558515

"Financial innovation and financial inclusion nexus in South Asian

Countries: Evidence from Symmetric and Asymmetric Panel investigation."

This paper could be interesting for the researchers in the area of “open economy macroeconomics and Economic growth”.  The paper simply studies relationship between financial inclusion and innovation a case study of six South Asian countries.  The study finds that financial inclusion and innovation can help developing a better financial sector.  Although the empirical section uses a different sample of dataset and countries, the methodology of the present study does not seem original and the results are convincing enough.

Major Suggestions:

Since this is an empirical paper, I suggest author(s) should include a brief descriptive statistics. I believe some readers (like me) may be curious to see these along with the presented results.

There are too many references included in the study back to back. It’s confusing and distracting.  I would rather see these references in a footnote or a separate table prepared if preferred.  

An appendix table for each variable with description and data sources would be better.

Conclusion section is not flowing with the paper. It has too many references and a tad bit long for a conclusion section.  Either divide into two or three parts or add some parts into previous sections.  A clean and with few references would be better for concluding remarks.  We can learn more from a complete and concise message.

Not sure with the abstract, feels like it needs to be rewritten, especially last couple of sentences.

Minor Suggestions:

Paper (table too) can be organized better overall to be much representable. Especially country list and associated time length.  I suggest author(s) to check with the previous published articles in ECPA and use similar formatting to represent their results.  It would be better to make them easier and clear for the readers.

Be careful with references too some or not completely checked.

Page 8, name of a reference repeated. Page 25, Chinese character reference included. Not sure if it’s included in the references section or not.

It seems to me that the paper is not ready yet. It’s rather rushed and not too organized.

Author Response

Dear reviewer,

                Thank you so much for your meaningful and thoughtful comments and insightful suggestion for further development. During the revised period, we tried our best in incorporating your suggestions and recommendation so that we can uplift your expectation.

                Please, still if you feel we need to do more work for further improvement, we love to do that, let us know.

Thank you so much

Regards

Major Suggestions:

Since this is an empirical paper, I suggest the author(s) should include a brief descriptive statistics. I believe some readers (like me) may be curious to see these along with the presented results.

 Response: Dear reviewer, by following your suggestion a brief descriptive statistics table inserted, please see Table-1

There are too many references included in the study back to back. It’s confusing and distracting.  I would rather see these references in a footnote or a separate table prepared if preferred.  

Response: Dear reviewer, thank you so much of your concern, but all the references are inserted in the text are directly related to the current research focused. Therefore, we do feel that it may produce more value if inserted in the text rather in the footnote.

An appendix table for each variable with description and data sources would be better.

Response: Dear reviewer, table inserted as per your instruction. Please see Appendix-B

The conclusion section is not flowing with the paper. It has too many references and a tad bit long for a conclusion section.  Either divide into two or three parts or add some parts into previous sections.  A clean and with few references would be better for concluding remarks.  We can learn more from a complete and concise message.

Response: Dear reviewer, please check the conclusion section. It is revised in accordance with your suggestions.

Not sure with the abstract feels like it needs to be rewritten, especially last couple of sentences.

Response: Dear reviewer, abstract of this article rewrites to make ease in reading.

Minor Suggestions:

Paper (table too) can be organized better overall to be much representable. Especially country list and associated time length.  I suggest the author(s) check with the previous published articles in ECPA and use similar formatting to represent their results.  It would be better to make them easier and clear for the readers.

Response: Dear reviewer, table formatted

Page 8, name of a reference repeated. Page 25, Chinese character reference included. Not sure if it’s included in the references section or not.

Response: all the references are rectified

Round 2

Reviewer 1 Report

no comment for author